# Updating the Classification of Chronic Inflammatory Enteropathies in Dogs

**DOI:** 10.3390/ani14050681

**Published:** 2024-02-21

**Authors:** Noémie Dupouy-Manescau, Tristan Méric, Odile Sénécat, Amandine Drut, Suzy Valentin, Rodolfo Oliveira Leal, Juan Hernandez

**Affiliations:** 1Oniris VetAgroBio Nantes, Department of Clinical Sciences, Nantes-Atlantic College of Veterinary Medicine and Food Sciences, 44300 Nantes, France; noemie.dupouy-manescau@oniris-nantes.fr (N.D.-M.); tristan.meric@oniris-nantes.fr (T.M.); odile.senecat@oniris-nantes.fr (O.S.); amandine.drut@oniris-nantes.fr (A.D.); 2Microbiota Interaction with Human and Animal Team (MIHA), Micalis Institute, AgrosParisTech, Université Paris-Saclay, Institut National de Recherche pour l’Agriculture, l’Alimentation et l’Environnement, 78350 Jouy-en-Josas, France; 3Hopia, Bozon Veterinary Clinic, 78280 Guyancourt, France; suzyvalentin@yahoo.fr; 4Associate Laboratory for Animal and Veterinary Sciences, AL4AnimalS, CIISA—Centre for Interdisciplinary Research in Animal Health, Faculty of Veterinary Medicine, University of Lisbon, 1649-004 Lisbon, Portugal; rleal@fmv.ulisboa.pt

**Keywords:** chronic inflammatory enteropathies, dogs, classification, dysbiosis, microbiota modulation

## Abstract

**Simple Summary:**

Chronic inflammatory enteropathies (CIEs) in dogs are currently classified according to clinical response to sequential treatment trials. The resulting recognized categories are food-responsive (FREs), antibiotic-responsive (AREs), immunosuppressant-responsive (IREs) and non-responsive enteropathies (NREs). Although this classification has benefited clinicians by providing a standardized approach to managing CIEs for almost a decade, the results of recent research challenge our understanding of the underlying pathophysiology and encourage a revision of these categories. The role of diet has been reinforced, and the gut microbiota has been acknowledged as an essential player in intestinal inflammation. Using antibiotics has been shown to result in deleterious, long-lasting effects; thus, approaches aimed at restoring a diverse and functional microbiota (prebiotics, probiotics, fecal microbiota transplantation, etc.) are required. We subsequently propose updating the classification of CIEs by replacing AREs with microbiota-related modulation-responsive enteropathies (MrMREs). The introduction of such a category can serve as a basis for further studies to assess the performance of options targeting the rebalance of the gut microbiota.

**Abstract:**

Chronic inflammatory enteropathies (CIEs) in dogs are currently classified based on response to sequential treatment trials into food-responsive (FREs); antibiotic-responsive (AREs); immunosuppressant-responsive (IREs); and non-responsive enteropathies (NREs). Recent studies have reported that a proportion of NRE dogs ultimately respond to further dietary trials and are subsequently misclassified. The FRE subset among CIEs is therefore probably underestimated. Moreover, alterations in the gut microbiota composition and function (dysbiosis) have been shown to be involved in CIE pathogenesis in recent research on dogs. Metronidazole and other antibiotics that have been used for decades for dogs with AREs have been demonstrated to result in increased antimicrobial resistance and deleterious effects on the gut microbiota. As a consequence, the clinical approach to CIEs has evolved in recent years toward the gradual abandonment of the use of antibiotics and their replacement by other treatments with the aim of restoring a diverse and functional gut microbiota. We propose here to refine the classification of canine CIEs by replacing the AREs category with a microbiota-related modulation-responsive enteropathies (MrMREs) category.

## 1. Introduction

Canine chronic inflammatory enteropathies (CIEs) are a group of diseases resulting in chronic (i.e., a 3-week duration or longer) or recurrent gastrointestinal clinical signs, including diarrhea, vomiting, nausea, borborygmus, flatulence, eructation, abdominal pain, weight loss or a combination of these signs. The diagnosis is made after the exclusion of extra-digestive disorders causing gastrointestinal clinical signs, intestinal parasitosis, and digestive neoplastic and infectious disorders [1].

The origin of the inflammation remains debated, but the current prevailing hypothesis is that the immune system excessively responds to environmental triggers (including to food and microbiota) in genetically predisposed individuals. Canine chronic inflammatory enteropathies are multifactorial diseases featuring a chronic immune response [2], the disruption of intestinal permeability [3], and the altered composition and function of the gut microbiota, referred to as dysbiosis [4]. Canine chronic inflammatory enteropathies are thought to arise in genetically prone individuals under the influence of epigenetic and environmental factors that alter immunotolerance and trigger the excessive activation of the innate and adaptive response of the host immunity. The microbiota–gut–brain axis is known for its bidirectional interactions between the central nervous system and the digestive tract and is gaining interest in regards to the onset of inflammatory bowel diseases [5].

CIEs are currently classified according to clinical response to treatment as food-responsive enteropathies (FREs), antibiotic-responsive enteropathies (AREs), immunosuppressant-responsive enteropathies (IREs) and non-responsive or refractory enteropathies (NREs). Inflammatory bowel diseases (IBDs) encompass both IREs and NREs with demonstrated mucosal inflammation. An additional group of CIEs named protein-losing enteropathies (PLEs) refers to all chronic enteropathies that result in hypoalbuminemia [6,7].

This classification has benefited clinicians by providing a standardized and consistent approach. However, our knowledge has continued to evolve, and significant limitations now make this system less suitable. Our objective was to propose an updated classification that considers recent advances in canine gastroenterology. With this review, we aim to briefly summarize the rationale of the current classification, discuss its limitations as underlined by recent research and propose a refined classification.

## 2. Current Classification of CIEs

The true prevalence of CIEs in dogs is unknown. In referral hospitals, CIEs account for 1–2% of cases, and these percentages are likely to be underestimated because about 10 to 20% of consultations undertaken in primary facilities relate to gastrointestinal clinical signs [1,6,8].

In dogs with chronic gastrointestinal signs, diagnoses of CIEs are considered after the exclusion of intestinal parasitism and extra-digestive disorders. In the current classification, once a CIE is suspected, several dietary trials are performed, usually including a highly digestible diet, a hydrolyzed protein diet, a fiber-rich diet, and/or a home-cooked novel protein diet with limited ingredients [9,10]. If clinical signs resolve with a dietary trial, the diagnosis falls in the FRE category. If clinical signs fail to improve, antibiotics have historically been advocated for (mainly metronidazole or tylosin). If the clinical response to the antibiotic trial is adequate, it is classified as an ARE. In the absence of a response to antibiotics, an ultrasound examination is usually pursued to exclude any focal disease of the gastrointestinal tract. Then, a histological examination of gastrointestinal biopsies is required to confirm mucosal inflammation and rule out diffuse neoplastic disorders and atypical infections. The dogs are then given an immunosuppressant, and the enteropathy is assigned to the IBD group (IRE or NRE, depending on the observed response). A graphical representation of the currently used categories of CIEs is provided in Figure 1 [1].

To date, no pathognomonic clinicals signs or clear scoring values have been determined to discriminate CIE categories.

Food-responsive enteropathies account for the majority of CIEs in dogs, ranging from 50 to 65% of cases [1,6,11,12]. Dogs with FREs seem to be younger than dogs with IREs (FRE median: 3 years, range: 0–12; IRE median: 6 years, range: 1–13 years, with *p* ≤ 0.001 in a cohort of 203 dogs diagnosed with CIEs [12]) and to exhibit more clinical signs consistent with large bowel disease. They have a lower canine chronic enteropathy clinical activity index (CCECAI) score in comparison with that of dogs suffering from AREs or IBD (median FRE: 6, range: 2–12; median ARE: 8, range: 0–14; median IRE: 9, range: 5–14, *p* ≤ 0.001 in a cohort of 203 dogs diagnosed with CIEs) [9,12,13,14]. However, no useful cut-off values for age or CCECAI scores and no specific clinical signs have been identified to help predict treatment response on an individual basis. Gluten-sensitive enteropathy in Irish setters and paroxysmal gluten-sensitive dyskesia in Border terriers (50% of affected dogs express gastrointestinal signs) are included in the FRE category [15,16].

Dogs with AREs enter clinical remission following metronidazole, tetracycline or tylosin administration. AREs account for 15 to 35% of CIEs [1,6,12,17]. Hypothesized underlying mechanisms include a decrease in the amount of deleterious bacteria within the gut microbiota and anecdotally direct immunomodulatory properties [18]. In particular, metronidazole and tylosin have recognized anti-inflammatory effects through modulating the synthesis of several mediators and cytokines [19,20]. Affected animals seem to be young large-breed dogs, with an over-representation of German shepherds [12,21]. However, age and dog breeds largely overlap among subtypes of CIEs. Relapses are frequent (diarrhea relapses occurred in 12 of 14 dogs within 30 days after tylosin discontinuation in a prospective study) [6,22].

Immunosuppressant-responsive enteropathies respond to glucocorticoids (prednisolone, budesonide), other immunosuppressants (cyclosporine, azathioprine, chlorambucil) or a combination of these treatments. IREs account for 10 to 25% of CIEs [6].

Five to forty-five percent of dogs with CIEs remain non-responsive (NREs) [6]. These CIEs carry a worse long-term prognosis with a high rate of euthanasia [9,23].

Protein-losing enteropathies (PLEs) encompass all enteropathies that potentiate protein leakage and malabsorption, resulting in hypoalbuminemia. Inflammatory PLEs are considered to be severe versions of CIEs. The usual diagnostic workup and therapeutic approach for CIEs are often modified in the case of hypoalbuminemia. Gut biopsies are usually taken immediately for histological examination and, instead of using a sequential treatment approach, immunosuppressants (mostly glucocorticoids and chlorambucil) are often started early, in combination with a highly digestible, ultra-low-fat diet, with or without antibiotics [1,24,25,26,27]. In addition, PLEs are frequently complicated with a hypercoagulable state and consequently require thromboprophylactic treatment [28]. Although the mechanism has not been clarified yet, the decrease in plasma antithrombin activity, the underlying disease as well as drug treatments (glucocorticoids) are conceivably contributors, with pulmonary thromboembolism being one of the most serious complications in these cases [29,30].

## 3. Limitations and Reconsideration of the Current Classification

### 3.1. Targeting the Microbiome toward Eubiosis

The role of the intestinal microbiota in health is now well established in humans and animals. Alterations in the gut microbiota composition and function (dysbiosis) are associated with canine CIEs. Essentially, there are overall decreased levels of richness and diversity. Pseudomonadota and Actinomycetota phyla are overrepresented, and Bacteroidota, Bacillota and Fusobacteria phyla are less abundant in most affected dogs. Among the Pseudomonadota phylum, the abundance of the Enterobacteriaceae family is increased. The abundances of the Bacteroidota phylum, Paraprevotellaceae family and Porphyromonas genus are increased. Finally, among the Bacillota phylum, declines in the *Faecalibacterium* (a single species of the genus *F. prausnitzii*), *Blautia* and *Turicibacter* genera and *Peptacetobacter hiranonis* have been reported [18,31,32,33,34,35]. The dysbiosis index (DI) was developed based on a mathematical algorithm using quantitative PCR testing of eight bacterial groups (i.e., *Blautia*, *Peptacetobacter hiranonis*, *Escherichia coli*, *Faecalibacterium*, *Fusobacterium*, *Streptococcus*, *Turicibacter* and total bacteria) that are commonly altered in dogs with CIEs [36]. A DI with a threshold value of 0 distinguishes dogs with chronic inflammatory enteropathy from healthy dogs with 74% sensitivity and 95% specificity. This tool makes it possible to detect dysbiosis and to monitor the return to eubiosis after appropriate treatment.

Gut dysbiosis entails the disruption of microbial-related metabolic pathways, such as short-chain fatty acids (SCFAs) and indole synthesis, bile acid (BA) biotransformation and proteolytic activities [31,37,38,39,40,41,42,43,44,45,46]. As an example, SCFAs (i.e., acetate, propionate and butyrate) are the main end products of the intestinal bacterial fermentation of non-digestible food components, such as dietary fiber. SCFAs are an essential energy source for colonocytes; they enhance epithelial barrier function by strengthening tight junctions and regulate T-lymphocyte function [47,48,49,50]. In another well-documented example, the deconjugation and dehydroxylation of primary BAs into secondary BAs are executed using enzymes carried by bacteria of the intestinal microbiota (*Bacteroides*, *Clostridium*, *Bifidobacterium*, *Lactobacillus*…) [51]. A decreased excretion of secondary BAs was detected in dogs with CIEs and presumably attributed to the decreased abundance of *Fusobacterium* and *Peptacetobacter hiranonis* [52]. Bile acids have been identified as antibacterial agents regulating gut microbial populations and as signaling ligands for multiple receptors (hormone farnesoid X receptor, Takeda G protein receptor 5…), influencing both host metabolism and immune response [51].

These data provide some evidence that intestinal dysbiosis promotes intestinal inflammation and thus provide the rationale to promote therapeutic strategies aimed at restoring eubiosis [40].

### 3.2. Towards Restricted Use of Antibiotics

For decades, metronidazole and other antibiotics have been successfully used in dogs with chronic gastrointestinal signs. However, evidence of deleterious effects on the gut microbiota is accumulating. Persistent changes in microbiota richness and composition are reported after metronidazole and tylosin treatments and may explain the frequency of clinical relapse after treatment interruption [4,12,22,53,54,55,56,57,58,59]. The design and conclusions of published papers studying the microbial consequences of using metronidazole or tylosin are summarized in Table 1.

In addition, there is a growing concern about antimicrobial resistance, which is one of the most serious and imminent One Health-related problems worldwide. A report showed that 54% of isolates of *Clostridium perfringens* from pet dogs with acute diarrhea had decreased susceptibility to metronidazole, including dogs not having had any previous treatment with antibiotics, suggesting that resistant strains might be transmitted from one individual to another [65]. As a consequence, dogs are considered a possible reservoir for antibiotic-resistant bacterial strains [66,67,68,69,70].

As a result, the clinical approach to CIEs has evolved in recent years toward the gradual abandonment of the use of antibiotics and their replacement by approaches aiming at restoring a functional gut microbiota [71].

### 3.3. Restoring Gut Microbiota

Several strategies aiming at restoring the functionality of the dysbiotic gut microbiota have been explored in recent years and include dietary changes; the use of prebiotics, probiotics, symbiotics, postbiotics; and fecal microbiota transplantation.

Prebiotics are non-viable substrates that serve as nutrients for beneficial microorganisms. They impact the composition of bacterial communities as well as microbial metabolic activities, including the synthesis of SCFAs [72]. Studies in rodent models and in humans suffering from IBD have demonstrated the benefits of prebiotic use which reduce histological lesions, proinflammatory cytokines and oxidative stress [73,74,75,76,77,78,79]. Although the overall level of evidence of prebiotics’ efficacy in humans with IBD is low, the results of two randomized clinical trials (RCTs) are of interest [80,81]. They evaluated germinated barley foodstuff (GBF), a dietary fiber classified as a prebiotic, which was demonstrated to lower clinical disease activity scores. In dogs with CIEs, two RCTs have been conducted to assess the potential benefits of prebiotics [82,83]. In the first RCT, β-glucans and mannan oligosaccharides (MOSs) did not provide any significant clinical benefits in dogs suffering from a CIE; however, the study was underpowered (with nine dogs in the supplemented group and ten dogs in the placebo group) [82]. In the second RCT, dogs with chronic diarrhea responding to a hydrolyzed diet (FRE) were randomly allocated either to a group supplemented with β-glucans, MOSs, chondroitin sulphate and glycosaminoglycans or to a placebo group. No significant differences in their relapse rates after returning to their initial diet were shown between the groups, but the study was also underpowered (with eight dogs in the test group vs. five in the control group) [83]. In a more recent RCT, a symbiotic-IgY supplement (probiotics: *Lactobacillus acidophilus*‚ *Lactobacillus casei*‚ *Enterococcus faecium* and *Bacillus subtilis*; prebiotics: beta-glucans, MOSs and D-mannose; immunoglobulin IgY derived from chicken egg yolk) led to decreased levels of fecal calprotectin and serum C-reactive protein; increased numbers of colonic mucosal *Clostridia* (class of *Bacillota* phylum) and *Bacteroides* (genus of *Bacteroidota* phylum); and decreased numbers of *Enterobacteriaceae* in CIE dogs. No clinical benefits were demonstrated in that study [84]. Therefore, evidence of any benefit does not exist in dogs and remains low in humans; nevertheless, no adverse effect has been reported in either species. Fructooligosaccharide (FOS) supplementation may be a more promising strategy and worth exploring [85,86].Probiotics are defined by the World Health Organization as “live microorganisms which when administered in adequate amounts confer a health benefit on the host” [87]. In dogs suffering from CIEs, RCTs investigating the benefit of probiotics remain scarce [84,88]. The multi-strain probiotic VSL#3 was compared to prednisolone/metronidazole in dogs suffering from IBD [89]. Significant improvement in CCECAI scores was found in both groups on day 90 when compared to those on day 0. In another controlled trial, dogs with IBD were administered conventional treatment (hydrolyzed protein diet, prednisolone and antibiotic) alone or in combination with *Saccharomyces boulardii* (10^9^/kg BID). The use of the yeast was associated with a lower CCECAI score on days 45 and 60 [90]. Although the evidence remains sparse, these data support the use of probiotics in dogs with CIEs. Based on individual experience and on the VSL#3 study, multi-strain probiotics should be considered.Symbiotics result from the combination of pre- and probiotics in the same product and have recently gained popularity for use in dogs and cats. Although studies about their efficacy are scarce, they seem to be promising options in terms of compliance [84].Postbiotics refer to dead microorganisms or microbial metabolic products beneficial to gut health [91]. Although postbiotics do not include live microorganisms, they may have beneficial properties based on their pleiotropic effects, including anti-inflammatory, antioxidant, immunomodulatory and anticancer properties [92,93]. After an exhaustive review of the literature, we found no study investigating the use of a product with only postbiotic properties in dogs with CIEs.Fecal microbiota transplantation (FMT) is now recognized as the standard of care in people suffering from recurrent *Clostridioides difficile* infection [94]. Available data suggest beneficial effects of FMT in patients with mild to moderate ulcerative colitis (UC), but there is insufficient evidence to recommend this therapeutic modality in routine clinical practice, and its use is currently limited to a research setting [95]. In dogs, only one RCT, one prospective study and six case reports/series on the use of FMT in dogs with CIEs/IBD have been published to date [96,97,98,99,100,101,102,103]. In the RCT, dogs with IBD were given either a FMT or a sterile saline enema as a placebo in addition to prednisolone and a hydrolyzed diet [102]. CCECAI scores significantly improved in both groups, but there were no significant differences between groups. The study was, however, underpowered (with seven dogs in the FMT group and six dogs in the control group) and might benefit from being repeated with larger cohorts of dogs. No adverse effects were observed in the dogs that received FMTs. The largest study is a retrospective case series on 41 dogs with CIEs not responding to diet, probiotics or immunosuppression [101]. The included dogs received one to five FMTs with fresh frozen feces via rectal enemas. In 31/41 dogs (76%), FMT was associated with clinical improvement. In 20/41 dogs, the dose of corticosteroids could be decreased and the antibiotics treatment was interrupted. The canine inflammatory bowel disease activity index (CIBDAI) significantly decreased. The study, however, did not include a control group to better evaluate the role played by FMT in the observed improvement [101]. As in humans, available data might support the use of FMT in dogs with CIEs, although evidence remains limited and it requires further assessment in a research setting.Interestingly, a recent publication reports clinical remission in two dogs suffering from NREs using cholestyramine, a bile acid sequestrant. The beneficial effects are suspected to be due to the correction of excess primary bile acids resulting from bile salt dysmetabolism associated with intestinal dysbiosis (a reduction in bacteria carrying the bile salt dehydrogenase) or due to the reduction in apical sodium-dependent bile acid transporter (ASBT) receptor expression at the brush border of the ileum in dogs with CIEs [104].

Table 2 lists relevant studies on strategies for modulating the gut microbiota in dogs with CIEs.

### 3.4. Reaffirming the Importance of Diet

Dietary changes remain the simplest way to modulate the intestinal microbiota. In addition to fiber-enriched foods that act as prebiotics, foods with hydrolyzed proteins, initially designed to reduce immunogenicity, have also shown favorable effects on the composition of the gut microbiota and on the biotransformation of bile salts [40]. It is therefore hypothesized that the clinical improvement observed with hydrolyzed diets is not exclusively linked to their immune effect but also to their beneficial effects on the gut microbiota.

Several case series reinforce the role of dietary changes by showing that cases initially classified as NREs could be ultimately reclassified as FREs after an additional dietary modification. A prospective study reported that clinical remission was achieved in eight out of ten dogs with steroid-resistant inflammatory PLEs using an additional dietary trial as a sole treatment change [105]. Another retrospective multicentric study, conducted on 142 dogs suffering from CIEs, initially classified 18% of cases as NREs. However, 88% of these “NREs” were then challenged by a novel commercial or home-cooked diet, and 68% of them responded [106]. It then appears that, although acknowledged as the first step of the current treatment-based approach, dietary benefit is not fully investigated in some dogs before moving toward other therapies. This might have resulted from discouragement from owners due to repeated non-responding trials. The recent evidence about “food-responding NREs” encourages owners to pursue at least one diet of each of the following categories for 2 weeks: a commercial highly digestible diet, commercial novel animal protein or hydrolyzed diet, home-cooked limited-ingredient diet and home-cooked novel protein diet. Also, evidence suggests that clinicians should retry dietary trials in the case of CIEs not responding properly to immunosuppression.

In addition, two recent publications have suggested that at least some inflammatory PLEs respond to diet only. In a case series of eleven Yorkshire terriers suffering from confirmed (*n* = 4) or presumed (*n* = 7) inflammatory PLEs that were all treated with a diet change without an immunosuppressant, clinical and biological improvements were observed in eight dogs [25]. In another case series of 27 dogs with confirmed inflammatory PLEs that were treated with an ultra-low-fat diet alone, 23 dogs improved (complete response in 12/27 dogs and partial response in 11/27 dogs) [107]. Interestingly, a CCECAI cut-off of eight showed a sensitivity of 83% and a specificity of 89% to discriminate between food-responders and non-food-responders. An ultra-low-fat diet might therefore be useful as a first-line treatment in many dogs suffering from inflammatory PLEs of low to moderate clinical severity. Based on the response to treatment, PLEs may be subcategorized as either food-responsive PLEs (FR-PLE) or non-food-responsive-PLEs requiring additional immunosuppressant treatment and then subclassified as immunosuppressant-responsive PLEs (IR-PLE) or non-responsive PLEs (NR-PLE).

## 4. Proposition of a Refined Classification of CIEs in Dogs

A proposal for an updated classification of canine CIEs is depicted in Figure 2.

### 4.1. Strengthening the Positioning of FRE

Several arguments lead to the resizing of the category of FRE and the consideration that its proportion is certainly underestimated:Several studies report that a proportion of NREs are in reality unidentified FREs, which increases the proportion of FREs and reinforces the idea that testing dietary changes constitutes the cornerstone of the diagnostic and therapeutic approach to chronic digestive disorders in dogs [105,106].Intestinal dysbiosis has emerged as an important contributing factor in intestinal inflammation in humans and animals. With diet being the most effective means of modulating the intestinal microbiota, it is likely that part of the beneficial effects observed clinically are due to the rebalancing of the gut microbiota.Finally, a proportion of PLEs respond to dietary changes and therefore fall into the FRE category and may be subcategorized as FR-PLE. In practice, it is always difficult to limit treatment to dietary changes in a potentially unstable animal. The CCECAI score could be a valuable tool to distinguish cases that could benefit from dietary changes alone (CCECAI < 8) from those also requiring the administration of an immunosuppressive agent (CCECAI > 8) [107].

### 4.2. Replacement of AREs by MrMREs

The use of antibiotics in the treatment of CIEs has been gradually replaced by other approaches to intestinal dysbiosis treatment. The term idiopathic intestinal dysbiosis has been suggested by others but does not reflect the extent of the existing treatment options [108]. We propose the replacement of AREs by a category that encompasses all methods of modulation of the intestinal microbiota (microbiota-related modulation-responsive enteropathy—MrMRE). This category would include enteropathies responding to prebiotics, probiotics, postbiotics, symbiotics, fecal microbiota transplantation, bile acid sequestrants and certain dietary changes. In this concept, FRE and MrMRE partially overlap because changing diet might result in clinical improvement attributable to the modulation of the microbiota or another immune or non-immune mechanism. The MrMRE category has the advantage of being adaptable over time because it is reasonable to assume that new modalities will appear in the years to come.

There is no reliable diagnostic criterion for MrMRE category. The clinical relevance of using the dysbiosis index to predict response to microbiota modulation strategies has not yet been demonstrated. This tool remains of interest to monitor animals suffering from intestinal dysbiosis. The association of hypocobalaminemia, hyperfolatemia and normal serum trypsin-like immunoreactivity, although very insensitive, should raise our suspicion of gut dysbiosis (increased consumption of vitamin B12 and increased folate production by the deviated microbiota) [109]. Hypocobalaminemia remains an unspecific observation since it can occur in exocrine pancreatic insufficiency, ileal malabsorption or Imerslund–Gräsbeck syndrome [109].

### 4.3. Reduction in the Place of IRE and NRE

Due to the growing proportion occupied by FREs and MrMREs, IREs and NREs now appear to constitute a minority of CIEs. New epidemiological studies are needed to determine their real proportions. Also, new treatments (including bile salt sequestrant and specific pre-/pro-/postbiotics aimed at restoring a functional microbiota) that should emerge as beneficial options in subsets of diseased individuals might further lead to a reduction in the proportion of dogs suitable for immunosuppression or dogs that do not respond to any treatment.

The diagnosis of IRE and NRE remains complex because it is the culmination of a complete process of exclusion of all other causes including FRE and MrMRE. A study showed that a serum CRP concentration of 9.1 mg/L or greater distinguished dogs with IRE from dogs with FRE or ARE with a sensitivity of 72% and a specificity of 100% [110]. The use of CRP testing to identify IRE dogs may be clinically relevant, but further studies performed with the current classification on large cohorts are required to confirm this result.

## 5. Conclusions

In conclusion, we have proposed a refined classification of canine CIEs that takes into account recent advances in our understanding of these diseases. The most relevant new insights are that (i) dietary changes are thought to be more important; (ii) the decreasing use of antibiotics leaves room for several strategies for modulating the intestinal microbiota; and (iii) cases requiring the use of immunosuppressive treatment seem to be less frequent than currently assumed.

Microbiota-related modulation-responsive enteropathies (MrMREs) might encompass an important part of CIEs by focusing on addressing a key pathogenic element. However, the best way to rebalance the gut microbiota remains to be clarified. Future research on such options should aim at testing their ability to restore a functional microbiota.

## Figures and Tables

**Figure 1 animals-14-00681-f001:**
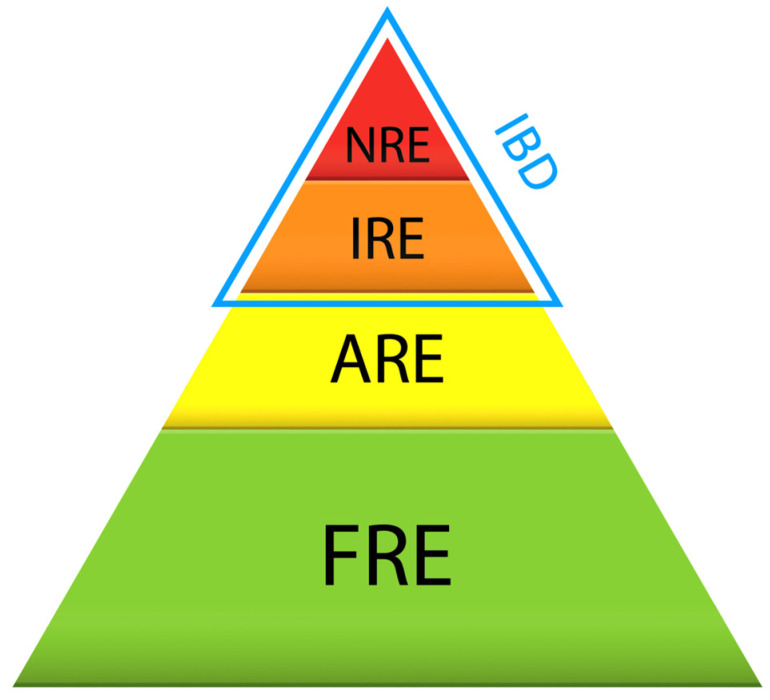
Representation of the “old” classification of canine chronic inflammatory enteropathies. FRE: food-responsive enteropathy, ARE: antibiotic-responsive enteropathy, IRE: immunosuppressant-responsive enteropathy, NRE: non-responsive enteropathy and IBD: inflammatory bowel disease if mucosal inflammation is demonstrated. Reprinted with permission from Ref. [1]. Copyright 2016 British Small Animal Veterinary Association.

**Figure 2 animals-14-00681-f002:**
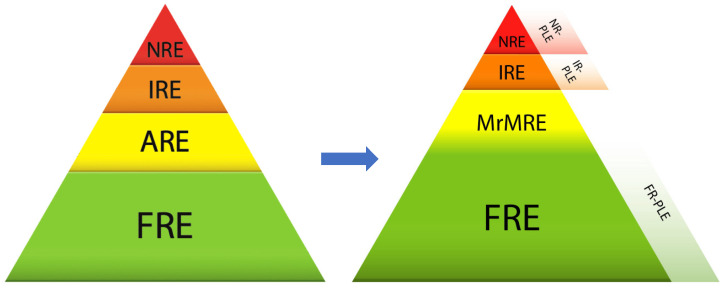
Proposal for a refined classification for chronic inflammatory enteropathies. FRE: food-responsive enteropathy, FR-PLE: food-responsive protein-losing enteropathy, MrMRE: microbiota-related modulation-responsive enteropathy, IRE: immunosuppressant-responsive enteropathy, IR-PLE: immunosuppressant-responsive protein-losing enteropathy, NRE: non-responsive enteropathy and NR-PLE: non-responsive protein-losing enteropathy.

**Table 1 animals-14-00681-t001:** Summary of studies assessing the impact of metronidazole and tylosin treatments on gut microbiota in dogs.

Antibiotic	Reference	Design	Conclusions
Metronidazole	[60]	Metronidazole was administered twice daily at 12.5 mg/kg PO to a group of five healthy research dogs and prednisolone at 1.0 mg/kg daily to a second group of five healthy research dogs for 14 days. Fecal samples were collected on days 0, 14, 28 and 42 (14 and 28 days after cessation).	In the group receiving metronidazole, their bacterial diversity indices significantly decreased on day 14 and recovered after cessation. Bacterial composition was also significantly altered by metronidazole on day 14 and returned to its initial proportions by day 42. Conversely, no effect of prednisolone was observed on either the bacterial diversity or composition.
	[61]	Prospective, non-randomized controlled study. Dogs fed various commercial diets were divided into three groups: the control group; the group receiving a hydrolyzed protein diet, followed by metronidazole administration; and the group receiving metronidazole only.	Metronidazole significantly changed microbiota composition in dogs treated with metronidazole only, and that change did not fully resolve 4 weeks after treatment discontinuation. Increased fecal total lactate and decreased fecal deoxycholic acid and lithocholic acid (secondary bile acids) were concurrently observed.
	[62]	Twelve healthy adult female dogs were used in an 8-week crossover design study. All dogs were fed a control diet during a 2-week baseline and then randomly allotted to one of the two treatment arms (diet only or diet + 1% prebiotic GNU100) for another 6 weeks. From weeks 2 to 4, dogs were orally administered metronidazole (20 mg/kg) twice daily.	Metronidazole reduced fecal microbial alpha diversity and *Blautia*, *Fusobacterium* (genus belonging to *Fusobacteria* phylum), *Turicibacter*, *Peptacetobacter hiranonis* and *Faecalibacterium* abundances and increased fecal *Streptococcus* (genus belonging to *Bacillota* phylum) and *Escherichia coli* (species belonging to *Pseudomonadota* phylum) abundances. Metronidazole also increased fecal primary bile acids and reduced secondary bile acid concentrations. Most changes returned to baseline by week 8.
Tylosin	[56]	Sixteen healthy dogs were randomized to receive 20 mg/kg of tylosin or a placebo capsule PO q12h for 7 days. The microbiota was assessed using 16S rRNA gene sequencing. Unconjugated bile acids were measured.	Samples from tylosin-exposed dogs exhibited decreased bacterial diversity characterized by a decrease in anaerobes *Fusobacteriaceae* (family of *Fusobacteria* phylum) and *Veillonellaceae* (family of *Bacillota* phylum) by day 7. Primary unconjugated bile acid fecal concentrations were increased on day 21 and day 63 compared to day 0 in dogs receiving tylosin. Changes did not uniformly resolve after discontinuation of tylosin on day 63.
	[63]	In vitro effects of tylosin, alone or associated with prebiotics, on a canine fecal suspension and the residue of in vitro digested dry dog food.	Tylosin resulted in lower total volatile fatty acids and *Lactobacillus* abundance; higher *Peptacedobacter* cluster I abundance after 6 h; and higher pH values, spermidine, and *E. coli* abundance throughout the study. When associated with tylosin, prebiotics counteracted some undesirable effects of the antibiotic treatment.
	[64]	Prospective, randomized, placebo-controlled study. Sixteen healthy dogs received 20 mg/kg PO tylosin once daily (days 1–7) and were randomly assigned to either receive one fecal microbiota transplantation (FMT) via enema (day 8), daily oral FMT capsules (days 8–21) or daily placebo capsules (days 8–21).	Tylosin altered the abundance of most evaluated bacteria and induced a significant decrease in secondary bile acid fecal concentrations by day 7 in all dogs. However, most parameters returned to baseline by day 14 in all dogs.
	[53]	Tylosin was administered at 20 to 22 mg/kg q 24 h PO for 14 days to five healthy dogs with a pre-existing jejunal fistula. Jejunal brush samples were collected through the fistula on days 0, 14 and 28.	Microbial diversity was reduced during tylosin treatment. On day 14, the proportions of *Enterococcus*-like organisms (genus of *Bacillota* phylum), *Pasteurella* spp. (genus of *Pseudomonadota* phylum) and *Dietzia* spp. (genus of *Actinomycetota* phylum) significantly increased, and proportions of *Spirochaetes* (class of *Spirochaetota* phylum), *Streptomycetaceae* (family of *Actinomycetota* phylum) and *Prevotellaceae* (family of *Bacteroidota* phylum) significantly decreased. On day 28, the proportion of *E. coli*-like organisms was increased in comparison to day 0, the phylogenetic composition of the microbiota was similar to that on day 0 in only two out of five dogs, and *Spirochaetes*, *Streptomycetaceae* and *Prevotellaceae* failed to recover.

**Table 2 animals-14-00681-t002:** Summary of studies assessing the effect of different strategies to modulate gut microbiota.

Strategy	Reference	Design	Conclusion
Prebiotics	[82]	Twenty-seven IBD dogs were randomized to be fed with chondroitin sulphate and prebiotics (resistant starch, β-glucans and mannaoligosaccharides) or placebo in addition to a hydrolyzed diet and were evaluated after 30, 60, 90 and 180 days of treatment.	No significant differences were found between groups at any point for CIBDAI, WSAVA histologic score or fecal microbiota evaluated by PCR-RLFP.
	[83]	Thirteen dogs with FREs were randomized to be fed a combination of prebiotics (β-glucans and mannan oligosaccharides), chondroitin sulphate and glycosaminoglycans or placebo in addition to a hydrolyzed diet for 10 weeks. Relapse rate was monitored every 2 weeks until week 18.	No significant differences were found over time or between groups for CCECAI, endoscopy scoring or histological scoring, nor in the relapse rate after switching back to the original diet.
Probiotics	[88]	A systematic review of clinical effect of probiotics in prevention or treatment of gastrointestinal disease in dogs, including twelve studies concerning acute gastrointestinal disease and five concerning chronic gastrointestinal disease.	The current data point toward a very limited and possibly clinically unimportant effect for prevention or treatment of acute gastrointestinal disease. For chronic gastrointestinal disease, dietary intervention remains the major key in treatment, whereas probiotic supplement seems not to add significant improvement.
	[89]	Twenty dogs with IBD were randomized to receive multi-strain probiotic (VSL#3) or prednisolone/metronidazole, monitored for 60 days and re-evaluated 30 days after completing treatment.	The CIBDAI and duodenal histology scores decreased between days 0 and 90 in both groups.
	[90]	Twenty dogs with CIEs were randomized to receive *Saccharomyces boulardii* (10^9^/kg BID) or a placebo, in addition to conventional treatment (hydrolyzed protein diet, prednisolone and antibiotic) for 60 days.	The administration of yeast was associated with a lower CCECAI score on days 45 and 60
Symbiotics	[84]	Twenty-four dogs with CIE were randomized to be fed a hydrolyzed diet and administered symbiotic-IgY (β-glucans, mannan oligosaccharides, D-mannose, *Lactobacillus acidophilusn*, *Lactobacillus casei*, *Enterococcus faecium*, *Bacillus subtilis* and immunoglobulin IgY derived from chicken egg yolk) or placebo for 6 weeks.	Dogs administered supplement exhibited decreased levels of fecal calprotectin and high-sensitivity C-reactive protein two weeks post-treatment, decreased levels of hs-CRP two- and six-weeks post-treatment, increased numbers of mucosal *Clostridia* and *Bacteroides* and decreased numbers of *Enterobacteriaceae* in colonic biopsies at the completion of the trial.
Fecal microbiota transplantation	[96]	Diversity analysis, differential abundance analysis and machine learning algorithms were applied to investigate the differences in microbiome composition between healthy and pre-FMT CIE-affected dogs, while CCECAI changes and microbial diversity metrics were used to evaluate oral freeze-dried fecal microbiota capsules’ effects.	In the healthy/pre-FMT comparison, differences were noted in alpha and beta diversity and a list of differentially abundant taxa was identified. Improvement of clinical signs was noted in 74% (20/27) of CIE-affected dogs, together with a decrease in CCECAI. Alpha and beta diversity variations between pre- and post-FMT were observed for each receiver, with a high heterogeneity in the response.
Fecal microbiota transplantation	[97]	A 10-year-old toy poodle diagnosed with IBD received nine FMTs by rectal enema within 6 months. 16S rRNA sequence analysis was performed before and after the FMTs.	Fecal microbiome diversity after FMT resembled that of the healthy donor dog’s fecal microbiome. The clinical symptoms improved remarkably with regard to the changes in the fecal microbiome. No observable side effects were noted.
	[98]	FMTs were performed in nine dogs with IBD. Fecal microbiome was examined via 16S rRNA sequencing in three dogs.	The proportion in *Fusobacteirum* in the post-FMT fecal microbiome was increased, and the CIBDAI decreased in all dogs.
	[99]	A 7-year-old Shiba dog diagnosed with protein-losing NRE received one FMT along with chlorambucil.	A single FMT via endoscopic procedure into the cecum and colon drastically recovered clinical signs and clinicopathological abnormalities and corrected dysbiosis in the dog. No recurrences or adverse events were observed.
	[100]	A 6-year-old Labrador dog diagnosed with IBD received FMT in the form of frozen oral capsules (five capsules/10 kg body weight for five consecutive days, along with prednisolone).	The CIBDAI switched from mild to clinically insignificant disease in 21 days. In the 18 months following FMT, the dog had some relapses defined as milder than before the FMT. No adverse effects were reported.
	[101]	Forty-one dogs with CIEs not responding to diet, probiotics or immunosuppression. Included dogs received one to five FMTs with fresh frozen feces via rectal enemas.	In 31/41 dogs (76%), FMT was associated with clinical improvement. In 20/41 dogs, the dose of corticosteroids was decreased and antibiotics were interrupted. The CIBDAI significantly decreased.
	[102]	Thirteen dogs with IBD were randomized to receive either FMT or placebo via rectal enema, along with cortisteroid therapy and a hypoallergenic diet, and were monitored for one month.	No significant differences in CCECAI between groups.
	[103]	Sixteen dogs with IBD received FMT, nine via an endoscopic procedure with five of them also given the transplant orally, and seven were administered by frozen capsules. They were monitored for 3 months. At the time of transplantation, all subjects were receiving immunosuppressants, antibiotics or both.	A clinical improvement was shown in most patients after transplantation, whether performed orally or endoscopically.
Bile acid sequestrants	[104]	One dog with NRE and one with IRE but with unacceptable corticosteroids side effects received cholestyramine (2 g q12–24 h).	Treatment with cholestyramine resulted in marked improvement of fecal consistency, frequency of defecation and activity level in both dogs.

## Data Availability

Not applicable.

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
