# Peer review of "Updating the Classification of Chronic Inflammatory Enteropathies in Dogs"

_animals, 2024, doi:10.3390/ani14050681_

Round 1

Reviewer 1 Report

Comments and Suggestions for Authors

Dear authors,

Your article is welcome, particularly at the current time of change in the paradigm of approach to chronic enteropathies in dogs.

However, in my opinion, the superficiality with which many subjects are covered leaves no room for readers, who are not specialized in gastroenterology, to benefit conveniently from your purpose.

Line 21 - The statement "and the gut microbiota has been recognized as an essential player in intestinal inflammation." leaves readers with more questions than helps them understand the phenomenon of intestinal inflammation.

It would be more useful to begin by defining the intestinal mucosal barrier as well as microbiota, including the notions of commensals and pathogens, and the acquisition of virulence factors in certain commensal strains in certain contexts, continuing to define dysbiosis and dysbiosis index.

Reference to the microbiota-gut-brain axis is essential, as is the notion of immunotolerance and the conditions for its alteration and its consequences (Lines 51 to 54).

Line 34 - "Metronidazole and other antibiotics that have been used for decades in dogs with AREs have been demonstrated to result in deleterious effects on gut microbiota." The harmful action of metronidazole and the consequences of the non-rational and widespread use of this drug go far beyond dysbiosis and this subject should be explored, even if briefly.

Although suggested in line 107, it is urgent to explain better, for example the anti-inflammatory activity of tylosin, which may have the most important role in the clinical resolution of cases of chronic enteropathy rather than its antibiotic capacity.

Line 124 - "In addition, PLEs are frequently complicated with a hypercoagulable state and consequently require thromboprophylactic treatment." It would be very interesting to find a justification for this event in the text given that most clinicians generally do not have this concept present.

Line 137 - "This entails the disruption of microbial-related metabolic pathways, such as short-chain fatty acids (SCFAs) and indole synthesis, bile acid biotransformation, and proteolytic activities." In my opinion, the opportunity to highlight the value of postbiotics in maintaining the integrity and as an energy source of intestinal cells should not be missed (not refered in lines 207 to 210), as well as the notion of primary bile acids and their transformation by specific microbiota agents into secondary ones (despite 233 to 238).

In my opinion, there seems to be a wrong strategy in referring to MOS instead of FOS, given that the former are much more suitable for reducing the microbial load in diarrhea caused by infectious agents, which does not fall within the scope of this work.

The part related to strategies to repair dysbiosis, which is the main leitmotif of this proposal, should be better organized and a table should be included with all relevant works.

Author Response

Dear Editor, 

We kindly thank you for the opportunity to peer-review and work our manuscript. We appreciate your time and consideration as well as of those of your team in pushing us to create the best possible article. We welcome the opportunity to summarize for you the changes. They are individually updated as a response to each of the reviewers.  Please, do not hesitate to contact us if you have any further suggestions.

We look forward to hearing from you,

Yours Sincerely,

Reviewer 1

Dear authors,

Your article is welcome, particularly at the current time of change in the paradigm of approach to chronic enteropathies in dogs.

Thank you for this kind comment.

However, in my opinion, the superficiality with which many subjects are covered leaves no room for readers, who are not specialized in gastroenterology, to benefit conveniently from your purpose.

Thank you for the comment. We have detailed certain sections in order to make the article understandable to as many people as possible.

Line 21 - The statement "and the gut microbiota has been recognized as an essential player in intestinal inflammation." leaves readers with more questions than helps them understand the phenomenon of intestinal inflammation.

This sentence being in the abstract, we do not have the space to detail it further here. However, in the text, we have detailed certain mechanisms that link the microbiota and intestinal inflammation.

It would be more useful to begin by defining the intestinal mucosal barrier as well as microbiota, including the notions of commensals and pathogens, and the acquisition of virulence factors in certain commensal strains in certain contexts, continuing to define dysbiosis and dysbiosis index.

Thank you for this recommendation. We clarified in the introduction the role of the unbalanced immune response, loss of epithelial permeability and intestinal dysbiosis in the origin of CIE (L52-55). To facilitate following the common thread, we have not addressed the notion of virulence factors of commensal bacteria because this only concerns a very specific entity that we do not address in this review, granulomatous colitis associated with E. coli. The dysbiosis index was introduced L 153-159.

Reference to the microbiota-gut-brain axis is essential, as is the notion of immunotolerance and the conditions for its alteration and its consequences (Lines 51 to 54).

Thank you for this recommendation. Reference to the microbiota-gut-brain axis and to immunotolerance was introduced L. 54-61

 Line 34 - "Metronidazole and other antibiotics that have been used for decades in dogs with AREs have been demonstrated to result in deleterious effects on gut microbiota." The harmful action of metronidazole and the consequences of the non-rational and widespread use of this drug go far beyond dysbiosis and this subject should be explored, even if briefly.

Thanks for your pertinent comment, we've expanded the abstract in this respect, and refer to it in section 3.1.

Although suggested in line 107, it is urgent to explain better, for example the anti-inflammatory activity of tylosin, which may have the most important role in the clinical resolution of cases of chronic enteropathy rather than its antibiotic capacity.

Your comment is fair, a paragraph enriching this point has been added in sectiob 3.1. L. 116-118.

Line 124 - "In addition, PLEs are frequently complicated with a hypercoagulable state and consequently require thromboprophylactic treatment." It would be very interesting to find a justification for this event in the text given that most clinicians generally do not have this concept present.

On your advice, we've added an explanation L. 136-140.

Line 137 - "This entails the disruption of microbial-related metabolic pathways, such as short-chain fatty acids (SCFAs) and indole synthesis, bile acid biotransformation, and proteolytic activities." In my opinion, the opportunity to highlight the value of postbiotics in maintaining the integrity and as an energy source of intestinal cells should not be missed (not refered in lines 207 to 210), as well as the notion of primary bile acids and their transformation by specific microbiota agents into secondary ones (despite 233 to 238).

Indeed, we have added detailed mechanisms L. 163-175 to enhance these points

In my opinion, there seems to be a wrong strategy in referring to MOS instead of FOS, given that the former are much more suitable for reducing the microbial load in diarrhea caused by infectious agents, which does not fall within the scope of this work.

We agree with this comment and added L. 231-233

The part related to strategies to repair dysbiosis, which is the main leitmotif of this proposal, should be better organized and a table should be included with all relevant works.

We thank the reviewer for this comment. We included a table with all relevant publication.

Reviewer 2 Report

Comments and Suggestions for Authors

The manuscript describes an attempt to introduce a more appropriate nomenclature hierarchy for IBC/CIE/CE in dogs. 

I think the manuscript is a great attempt to help categorize a difficult disease process but I do think it needs a little work. 

I do think diagnostic criteria needs to be included for each category, along with a substage. 

FRE should include gluten sensitivity in the border terriers with dyskinesia or Irish Setters as well. 

For example for for the FRE there is no test, but with the MMRE a dysbiosis index should be elevated, or folate or low cobalamin with normal pancreatic function, diarrhea panel with clostridium, salmonella, campylobacter or with IRE elevated CRP, calprotectin, S100A12 etc. and NRE should be non--responsive to anything 

The substage can be PLE, so FRE with PLE or MMRE with PLE. 

I do think bile acid diarrhea should fall under a category within MMRE and this should be potentially modified to microbiota or their metabolites modulation-responsive enteropathy—MmMRE or something similar. As bile acid changes one of the main changes seen with microbiota changes. 

Thank you and good luck!

Comments on the Quality of English Language

No major issues 

Author Response

The manuscript describes an attempt to introduce a more appropriate nomenclature hierarchy for IBC/CIE/CE in dogs. 

I think the manuscript is a great attempt to help categorize a difficult disease process but I do think it needs a little work. 

I do think diagnostic criteria needs to be included for each category, along with a substage. 

Accordingly, we included diagnostic criteria for each category, the use of available biomarkers and inserted substages in Fig. 2.

FRE should include gluten sensitivity in the border terriers with dyskinesia or Irish Setters as well. 

One sentence has been added in the text about that in the description of FRE category (L110-112).

For example for the FRE there is no test, but with the MMRE a dysbiosis index should be elevated, or folate or low cobalamin with normal pancreatic function, diarrhea panel with clostridium, salmonella, campylobacter or with IRE elevated CRP, calprotectin, S100A12 etc. and NRE should be non--responsive to anything 

  1. 419-427 + 436-441

The substage can be PLE, so FRE with PLE or MMRE with PLE. 

We modified Fig. 2 introducing substages (FR-PLE). To our knowledge, MrMRE with PLE is not described.

I do think bile acid diarrhea should fall under a category within MMRE and this should be potentially modified to microbiota or their metabolites modulation-responsive enteropathy—MmMRE or something similar. As bile acid changes one of the main changes seen with microbiota changes. 

Thank you for this proposal which led to a constructive debate between the authors. Broadening the name to include cholestyramine-responsive diarrhea is an excellent idea. We propose the name Microbiota related Modulation Responsive Enteropathy (MrMRE) for reasons of readability and pronunciation.

Thank you and good luck!

Thank you for your support

Round 2

Reviewer 1 Report

Comments and Suggestions for Authors

Dear authors,

In my opinion, the article has gained in scientific rigor and is much more solid, congratulations!

I will then note some additional points.

Keywords: chronic inflammatory enteropathies; dogs; classification; metronidazole; dysbiosis; microbiota modulation

Why highlight this active principle in this article, which is more related to the old paradigm than to what we are now proposing?

In my opinion, the paragraph between lines 52 and 55 does not conflict with the rest of the introduction and should be incorporated into it again.

Line 532 - deviated flora-microbiota

Line 532 - "Hypocobalaminemia remains an unspecific observation since it can occur in exocrine pancreatic insufficiency, ileal malabsorption or Imerslund-Gräsbeck syndrome.  Please add  a reference.

Table 2 - Why do they classify cholestyramine as a postbiotic?

Page 14 - On page 14 there is a figure without caption. Is it part of Figure 2?

Please review the bibliographic references, as in,...

108 - From Bench Top to Clinics: How New Tests Can Be Helpful in Diagnosis and Management of Dogs with Chronic Enteropathies. Juan Hernandez, Julien Rodolphe Samuel Dandrieux. Vet Clin North Am Small Anim Pract. 2021 Jan;51(1):137-153. doi: 10.1016/j.cvsm.2020.09.008.

Author Response

Dear authors,

In my opinion, the article has gained in scientific rigor and is much more solid, congratulations!

We kindly thank you for the opportunity to peer-review and work our manuscript. We appreciate your time and consideration in pushing us to create the best possible article.

I will then note some additional points.

Keywords: chronic inflammatory enteropathies; dogs; classification; metronidazole; dysbiosis; microbiota modulation

Why highlight this active principle in this article, which is more related to the old paradigm than to what we are now proposing?

Thank you for identifying this error. We have removed metronidazole from the list of keywords

In my opinion, the paragraph between lines 52 and 55 does not conflict with the rest of the introduction and should be incorporated into it again.

We reincorporated lines 52-55

Line 532 - deviated flora-microbiota

Adressed

Line 532 - "Hypocobalaminemia remains an unspecific observation since it can occur in exocrine pancreatic insufficiency, ileal malabsorption or Imerslund-Gräsbeck syndrome.  Please add  a reference.

Adressed

Table 2 - Why do they classify cholestyramine as a postbiotic?

We removed the postbiotic section because there was no clinical publication in dogs and created the “bile acid sequestrants” section.

Page 14 - On page 14 there is a figure without caption. Is it part of Figure 2?

We do not see a figure without a legend. Could this be a layout issue?

Please review the bibliographic references, as in,...

108 - From Bench Top to Clinics: How New Tests Can Be Helpful in Diagnosis and Management of Dogs with Chronic Enteropathies. Juan Hernandez, Julien Rodolphe Samuel Dandrieux. Vet Clin North Am Small Anim Pract. 2021 Jan;51(1):137-153. doi: 10.1016/j.cvsm.2020.09.008.

Error corrected. We also corrected an error in ref 106
